# Proton transfer pathway in anion channelrhodopsin-1

Masaki Tsujimura[1†], Keiichi Kojima[2†], Shiho Kawanishi[2], Yuki Sudo[2]*, Hiroshi Ishikita[1,3]*

[1]Department of Applied Chemistry, The University of Tokyo, Tokyo, Japan; [2]Graduate School of Medicine, Dentistry and Pharmaceutical Sciences, Okayama University, Okayama, Japan; [3]Research Center for Advanced Science and Technology, The University of Tokyo, Tokyo, Japan

**Abstract** Anion channelrhodopsin from *Guillardia theta* (*Gt*ACR1) has Asp234 (3.2 Å) and Glu68 (5.3 Å) near the protonated Schiff base. Here, we investigate mutant *Gt*ACR1s (e.g., E68Q/D234N) expressed in HEK293 cells. The influence of the acidic residues on the absorption wavelengths was also analyzed using a quantum mechanical/molecular mechanical approach. The calculated protonation pattern indicates that Asp234 is deprotonated and Glu68 is protonated in the original crystal structures. The D234E mutation and the E68Q/D234N mutation shorten and lengthen the measured and calculated absorption wavelengths, respectively, which suggests that Asp234 is deprotonated in the wild-type *Gt*ACR1. Molecular dynamics simulations show that upon mutation of deprotonated Asp234 to asparagine, deprotonated Glu68 reorients toward the Schiff base and the calculated absorption wavelength remains unchanged. The formation of the proton transfer pathway via Asp234 toward Glu68 and the disconnection of the anion conducting channel are likely a basis of the gating mechanism.

**\*For correspondence:**
sudo@okayama-u.ac.jp (YS);
hiro@appchem.t.u-tokyo.ac.jp
(HI)

†These authors contributed
equally to this work

**Competing interest:** The authors
declare that no competing
interests exist.

**Reviewing Editor:** Qiang Cui,
Boston University, United States

## Introduction

Anion channelrhodopsins (ACRs) are light-gated anion channels that undergo photoisomerization at the retinal chromophore, which is covalently attached to a conserved lysine residue via the protonated Schiff base, from all-*trans* to 13-*cis*. Natural ACRs were identified in the cryptophyte *Guillardia theta* (*Gt*ACR1 and *Gt*ACR2) (*Govorunova et al., 2015*). ACRs hyperpolarize the membrane through anion import and can widely be used as neural silencing tools in optogenetics (*Wiegert et al., 2017*; *Miyazaki et al., 2019*). Microbial rhodopsins have acidic residues or Cl⁻ at the Schiff base moiety to stabilize the protonated Schiff base as counterions. Counterions play a major role in determining the absorption wavelength and the function of the protein (*Tsujimura and Ishikita, 2020*). The X-ray crystal structures of *Gt*ACR1 show that two acidic residues, Glu68 and Asp234, exist at the corresponding positions (*Figure 1*; *Kim et al., 2018*; *Li et al., 2019*).

It was proposed that both Glu68 and Asp234 were protonated in *Gt*ACR1 (*Kim et al., 2018*; *Sineshchekov et al., 2016*; *Yi et al., 2016*; *Kandori, 2020*) in contrast to other microbial rhodopsins because the absorption wavelengths remain unchanged upon the E68Q and D234N mutations (*Kim et al., 2018*; *Sineshchekov et al., 2016*; *Yi et al., 2016*). Indeed, the C=C stretching frequency of the retinal is not significantly affected upon the E68Q and D234N mutations in resonance Raman spectroscopy, which implies that the electrostatic interaction between the retinal and protein environment remains unchanged (*Yi et al., 2016*). In addition, the C=O stretching frequency for a protonated carboxylate, which is observed in the wild-type *Gt*ACR1, disappears in the E68Q (*Yi et al., 2017*; *Dreier et al., 2021*) and D234N (*Kim et al., 2018*) *Gt*ACR1s according to Fourier transform infrared (FTIR) spectroscopy analysis.

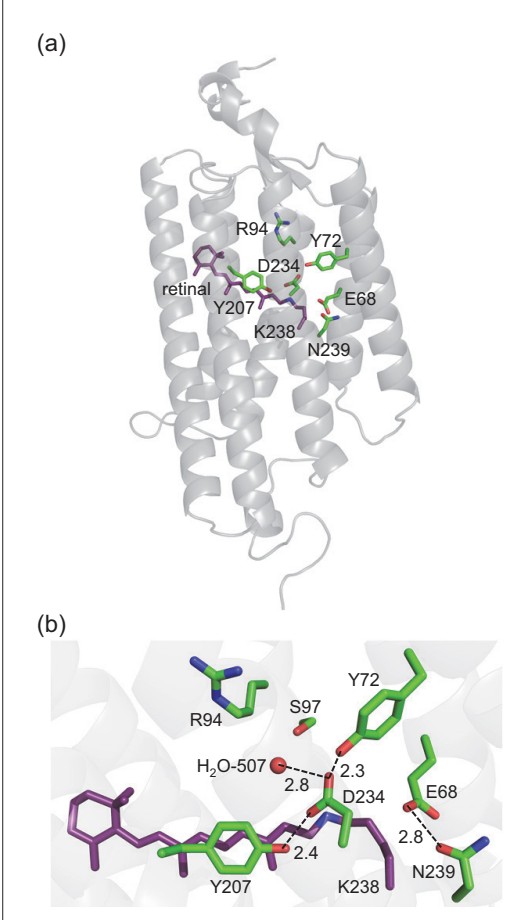

**Figure 1.** GtACR1. (**a**) Overview of the GtACR1 structure. (**b**) Residues near the Schiff base in GtACR1 (PDB code: 6EDQ; **Li et al., 2019**). Dotted lines indicate H-bonds.

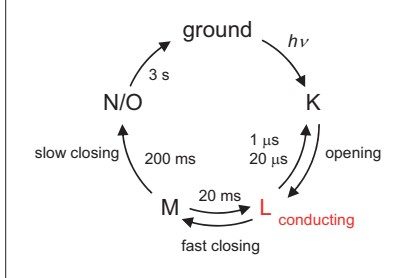

**Figure 2.** Photocycle of GtACR1 proposed in **Sineshchekov et al., 2016**. The L- to M-state and M- to N/O-state transitions involve deprotonation and reprotonation of the Schiff base, respectively.

The online version of this article includes the following figure supplement(s) for figure 2:

**Figure supplement 1.** Representative MDgenerated conformations (left panel) and the distances (d) between Asp234 and the adjacent residues during the 1 ns production run (right panel) in the wild-type GtACR1s with (**a**) protonated Glu68/deprotonated Asp234, (**b**) protonated Glu68/protonated Asp234, and (**c**) deprotonated Glu68/protonated Asp234.

Nevertheless, it is an open question whether Asp234 is protonated. Kim et al. pointed out that the loss of photocurrent in the D234N GtACR1 cannot be easily understood if Asp234 is protonated as the influence of the mutation of protonated aspartate to asparagine on the protein function is often small (**Kim et al., 2018**). GtACR1 crystal structures show that the residues at the Schiff base moiety are highly conserved between GtACR1 and bacteriorhodopsin (BR). Tyr57, Arg82, Tyr185, and Lys216, which are responsible for the low $pK_a$ of –2 for the counterion Asp212 in BR (**Saito et al., 2012**), are fully conserved as Tyr72, Arg94, Tyr207, and Lys238 in GtACR1. Note that the counterion Asp85 in BR, which increases $pK_a$(Asp212) by 6 (**Saito et al., 2012**), is replaced with Ser97 in GtACR1. This suggests that the $pK_a$ of Asp234 in GtACR1 is even lower than the low $pK_a$ of –2 for Asp212 in BR. According to Li et al., Tyr72 and Tyr207 donate H-bonds to Asp234 in GtACR1 (**Li et al., 2019**): this suggests that deprotonated Asp234 is stabilized, decreasing $pK_a$(Asp234), as observed for deprotonated Asp212 in BR. In addition, resonance Raman spectroscopy analysis indicates that the Schiff base Lys238 is also protonated (**Yi et al., 2016**) as observed in other microbial rhodopsins. The presence of the positively charged Schiff base needs to have an adjacent negative charge (e.g., deprotonated acidic residue) to effectively decrease the energy in GtACR1. To the best of our knowledge, microbial rhodopsins have more than one deprotonated acidic residue adjacent to the retinal Schiff base (e.g., **Tsujimura and Ishikita, 2020**). This also holds true for channelrhodopsin from *Chlamydomonas noctigama* (Chrimson) and rhodopsin phosphodiesterase (Rh-PDE), which have both deprotonated and protonated acidic residues near the Schiff base (**Vierock et al., 2017**; **Watari et al., 2019**). That is, either Glu68 or Asp234 may be deprotonated in GtACR1. Deprotonation of Asp234 is energetically more favorable than deprotonation of Glu68 in the presence of protonated Schiff base as the electrostatic interaction with Asp234 (3.2 Å) is larger than with Glu68 (5.3 Å). Alternatively, Cl⁻ may exist and act as a counterion, as observed in Cl⁻ pumping rhodopsins (**Kim et al., 2016**; **Kolbe et al., 2000**). However, the corresponding electron density is not observed in GtACR1 (**Kim et al., 2018**; **Li et al., 2019**). In addition, no spectral changes are reported upon deionization of

the sample or exchange from Cl⁻ to $SO_4^{2-}$ buffer (*Sineshchekov et al., 2016*). So far, the counterion of *Gt*ACR1 remains unknown.

Recently, Dreier et al. proposed that Asp234 is deprotonated in the dark and acts as a counterion according to FTIR measurements and molecular dynamics (MD) simulations (*Dreier et al., 2021*). The C=O stretching frequencies of 1740 (–)/1732 (+) cm⁻¹ for protonated Asp234 at 77 K observed by *Kim et al., 2018* were not observed at 293 K by *Dreier et al., 2021*. In addition, MD simulations indicated that the H-bond network that involves Tyr72, Tyr207, and Asp234 was stable with deprotonated Asp234 but unstable with protonated Asp234 (*Dreier et al., 2021*). Indeed, the presence of deprotonated Asp234 was already suggested based on the homology modeling of *Gt*ACR2 (*Kojima et al., 2018*) before the crystal structures of *Gt*ACR1 were reported.

*Gt*ACR1 undergoes a photocycle including K, L, M, N, and O intermediates (*Figure 2*; *Sineshchekov et al., 2016*). The L-state represents the anion conducting state. The L- to M-state transition involves the deprotonation of the Schiff base and the photocurrent decay. The fast photocurrent decay (fast channel closing) corresponds to the M-state formation (i.e., proton release from the Schiff base), and the slow photocurrent decay corresponds to the M-state decay (*Sineshchekov et al., 2016*; *Sineshchekov et al., 2015*; *Figure 2*). Glu68 is likely to accept a proton from the Schiff base upon the M-state formation as a decrease in the accumulation of the M-state was observed in the E68Q *Gt*ACR1 (*Sineshchekov et al., 2016*). However, it remains unclear whether Glu68 is the initial proton acceptor in the wild-type *Gt*ACR1. The *Gt*ACR1 crystal structures show that Asp234 is closer to the Schiff base (*Figure 1*; *Kim et al., 2018*; *Li et al., 2019*; *Li et al., 2021*). In addition, the E68Q mutation did not completely inhibit the M-state formation, which indicates that an alternative proton accepter exists (*Sineshchekov et al., 2016*).

To identify the counterion and clarify the proton-mediated gating mechanism of *Gt*ACR1, we investigate the Glu68 and Asp234 mutant proteins (E68Q, E68D, D234N, D234E, and E68Q/D234N) expressed in HEK293 cells. The protonation states are calculated by solving the Poisson–Boltzmann equation and evaluated by conducting MD simulations. Using a quantum mechanical/molecular mechanical (QM/MM) approach, the absorption wavelengths are calculated and the microscopic origin of the wavelength shifts upon the mutations is analyzed.

## Results

### Protonation states of Glu68 and Asp234

The protonation pattern (*Table 1*) and $pK_a$ values (*Table 2* and *Table 3*) calculated solving the linear Poisson-Boltzmann equation show that Asp234 is deprotonated, whereas Glu68 is protonated in the *Gt*ACR1 crystal structures (*Kim et al., 2018*; *Li et al., 2019*; *Table 1*; *Supplementary file 1A*). The calculated protonation pattern shows that Asp234 is deprotonated in the wild-type *Gt*ACR1 even using the MD-generated conformations with protonated Glu68/protonated Asp234 or deprotonated Glu68/protonated Asp234 (*Table 1*, *Supplementary file 1B*, *Figure 2—figure supplement 1*). These results suggest that deprotonation of Asp234, 'the only residue directly interacting with the protonated Schiff base (*Li et al., 2019*)', is a prerequisite to stabilize the protonated Schiff base, as suggested by *Dreier et al., 2021*.

$pK_a$(Asp234) = –5 (*Table 2*) is significantly low and even lower than $pK_a$(Asp212) = –2 in BR (*Saito et al., 2012*). The crystal structures show that the residues at the Schiff base moiety are highly conserved between *Gt*ACR1 and BR. Tyr72 and Tyr207 donate H-bonds to each carboxyl O site of Asp234 in *Gt*ACR1 (*Li et al., 2019*; *Dreier et al., 2021*), whereas Tyr57 and Tyr185 donate H-bonds to each carboxyl O site of Asp212 in BR (*Saito et al., 2012*). Thus, each tyrosine residue stabilizes the deprotonated state of Asp234, decreasing $pK_a$(Asp234) in *Gt*ACR1 by ~3 (*Table 2*), as observed in BR (*Saito et al., 2012*). The tendency is also observed for the conserved residue pairs, Arg94/Arg82 and Lys238/Lys216, in *Gt*ACR1/BR (*Table 2*). Asp85, which increases $pK_a$(Asp212) in BR by 6, is replaced with Ser97, which has no influence on $pK_a$(Asp234) in *Gt*ACR1 (*Table 2*). This discrepancy contributes to the low $pK_a$(Asp234) in *Gt*ACR1, which is lower than $pK_a$(Asp212) in BR. As far as the original geometry of the *Gt*ACR1 crystal structure is analyzed, no residue that increases $pK_a$(Asp234) significantly is identified (*Table 2*).

Recently, Li et al. reported the *Gt*ACR1 conformation (pre-activating state), where Arg94 forms a salt-bridge with Asp234 (*Li et al., 2021*). The influence of Arg94 on $pK_a$(Asp234) (~3) indicates that

**Table 1.** Protonation states of Glu68 and Asp234.
Changes in the protonation states are in bold. – indicates not applicable.

|  |  | Fixed state during MD* | Calculated state after MD† |
|---|---|---|---|
| Wild-type (1) | E68 | Protonated | Protonated |
|  | D234 | Deprotonated | Deprotonated |
| Wild-type (2) | E68 | Protonated | Protonated |
|  | D234 | Protonated | **Deprotonated** |
| Wild-type (3) | E68 | Deprotonated | **Protonated** |
|  | D234 | Protonated | **Deprotonated** |
| E68D (1) | D68 | Protonated | Protonated |
|  | D234 | Deprotonated | Deprotonated |
| E68D (2) | D68 | Deprotonated | **Protonated** |
|  | D234 | Deprotonated | Deprotonated |
| E68Q (1) | Q68 | – | – |
|  | D234 | Deprotonated | Deprotonated |
| E68Q (2) | Q68 | – | – |
|  | D234 | Protonated | **Deprotonated** |
| D234E (1) | E68 | Protonated | Protonated |
|  | E234 | Deprotonated | Deprotonated |
| D234E (2) | E68 | Protonated | Protonated ‡ |
|  | E234 | Protonated | Protonated ‡ |
| D234N (1) | E68 | Deprotonated | Deprotonated |
|  | N234 | – | – |
| D234N (2) | E68 | Protonated | Protonated § |
|  | N234 | – | – |
| E68Q/D234N | Q68 | – | – |
|  | N234 | – | – |

*The system was equilibrated for 5 ns. 10 conformations were sampled at 0.1 ns intervals during the 1 ns production run.

†Protonation patterns obtained using the MD-generated conformations.

‡Although we were able to obtain the MD-generated conformation with protonated Glu68 and protonated Glu234, which was confirmed in the calculated protonation pattern, the conformation cannot reproduce the experimentally measured absorption wavelength (**Supplementary file 1D**) and is unlikely relevant to the D234E *Gt*ACR1.

§Although we were able to obtain the MD-generated conformation with protonated Glu68, which was confirmed in the calculated protonation pattern, the protonation state is not consistent with deprotonated Glu68 suggested in FTIR studies by **Dreier et al., 2021**.

the electrostatic link between Arg94 and Asp234 exists even in the ground state (**Table 2**). It seems possible that the electrostatic interaction between deprotonated Asp234 and channel-gating Arg94 (**Table 2**) is absent in the D234N *Gt*ACR1, leading to the loss of the photocurrent (**Kim et al., 2018**).

In contrast, $pK_a$(Glu68) is high, 12 (**Table 3**), which is consistent with the reported protonation state of Glu68 (**Yi et al., 2016**; **Yi et al., 2017**; **Dreier et al., 2021**). The high $pK_a$(Glu68) value can be primarily due to the presence of anionic Asp234 whose deprotonated state is stabilized by the protonated Schiff base (**Table 3**). Charge neutral Ala53 exists at the corresponding position in BR, which is also consistent with the protonation of Glu68 (**Table 3**).

**Table 2.** p$K_a$ for Asp234 in *Gt*ACR1 and Asp212 in BR.

|  | *Gt*ACR1 |  | BR |  |
|---|---|---|---|---|
| p$K_a$ | Asp234 | −4.9 | Asp212 | −2.0* |
| | | | | |
| Contribution | | | | |
| | Schiff base | −9.7 | Schiff base | −8.7* |
| | Arg94 | −3.3 | Arg82 | −6.3* |
| | Tyr72 | −3.3 | Tyr57 | −3.4* |
| | Tyr207 | −3.4 | Tyr185 | −3.3* |
| | Ser97 | 0.5 | Asp85 | 6.0* |

*See **Saito et al., 2012**.

Exceptionally, Glu68 is deprotonated only in the D234N *Gt*ACR1. The influence of the protonated Schiff base is weaker on Glu68 than on Asp234 (**Table 2** and **Table 3**), which allows to stabilize the putative protonated Glu68 conformation in MD simulations (**Table 1**). However, the experimentally measured absorption wavelength cannot be reproduced unless Glu68 is deprotonated in the D234N *Gt*ACR1 (**Table 4**, **Supplementary file 1C**). This is consistent with the absence of the 1708 cm⁻¹ band in the D234N *Gt*ACR1, which is assigned to protonated Glu68 in FTIR measurements (**Dreier et al., 2021**). These results confirm that the presence of a negative charge at the protonated Schiff base moiety is a prerequisite to stabilize the protonated Schiff base, as observed in other microbial rhodopsins (**Tsujimura and Ishikita, 2020**), including Chrimson and Rh-PDE (**Vierock et al., 2017**; **Watari et al., 2019**).

## Absorption wavelengths of the Glu68 and Asp234 mutant *Gt*ACR1s

The experimentally measured absorption wavelengths for the Glu68 and Asp234 mutant proteins (E68Q, E68D, D234N, D234E, and E68Q/D234N) expressed in HEK293 cells are shown in **Figure 3**.

In the present study, the experimentally measured absorption wavelengths are the same for the wild-type and D234N *Gt*ACR1s (**Table 4**), which is consistent with results reported previously (**Kim et al., 2018**; **Sineshchekov et al., 2016**; **Yi et al., 2016**). Notably, the calculated absorption wavelengths are also the same for the wild-type and D234N *Gt*ACR1s irrespective of deprotonated Asp234 in the wild-type *Gt*ACR1 (**Table 4**, **Supplementary file 1C**, **Figure 3—figure supplements 1 and 2**). This can be explained as, in the D234N *Gt*ACR1, deprotonated Glu68 moves toward the positively charged Schiff base, which fully substitutes a role of deprotonated Asp234 in stabilizing the positively charged Schiff base (**Figure 4a and b**, **Table 5**). A similar conformation of Glu68, which orients toward the Schiff base, was previously reported for the corresponding residues of *Gt*ACR2 (Glu64) (**Kojima et al., 2018**) and channelrhodopsin from *Chlamydomonas reinhardtii* (Glu90) (**Volkov et al., 2017**). Thus, the absence of the change in the absorption wavelength upon D234N mutation (**Kim et al., 2018**) does not necessarily indicate that Asp234 is protonated in the wild-type *Gt*ACR1.

The existence of deprotonated Asp234 in the wild-type *Gt*ACR1 can also be understood from the absorption wavelength in the D234E *Gt*ACR1. The distance between Glu234 and the Schiff base (2.9 Å in MD-generated conformations) in the D234E *Gt*ACR1 is shorter than that between Asp234 and the Schiff base (3.4 Å in MD-generated conformations) in the wild-type *Gt*ACR1 because glutamate is longer than aspartate (**Figure 4a and c**, **Figure 4—figure supplement 1**). The absorption wavelength is short as the electrostatic interaction between the deprotonated counterion and the protonated Schiff base is strong (**Tsujimura and Ishikita, 2020**). Remarkably, the D234E mutation leads to a decrease in the absorption wavelength (**Table 4**, **Figure 3**, **Figure 3—figure supplements 1 and 2**), which suggests that Asp234 is deprotonated in the wild-type *Gt*ACR1. The decrease in the measured absorption wavelength of 10 nm could not be reproduced when we forced Asp234 in wild-type and Glu234 in D234E *Gt*ACR1s to protonate (**Supplementary file 1D**). The electrostatic contributions of charge-neutral protonated

**Table 3.** p$K_a$ for Glu68 in *Gt*ACR1.
The corresponding sites of BR are shown in the same line. – indicates not applicable.

|  | *Gt*ACR1 |  | BR |  |
|---|---|---|---|---|
| p$K_a$* | Glu68 | 12 | Ala53 | – |
| | | | | |
| Contribution | | | | |
| | Schiff base | −6.6 | Schiff base | – |
| | Arg94 | −1.3 | Arg82 | – |
| | Asn239 | 2.5 | Val217 | – |
| | Asp234 | 4.9 | Asp212 | – |

*See **Saito et al., 2012**.

**Table 4.** Calculated and experimentally measured absorption wavelengths $\lambda_{max}$ (nm).
$\Delta\lambda$ (nm) denotes the shift with respect to the wild-type *Gt*ACR1.

| | Calculated* | | Measured | |
|---|---|---|---|---|
| | $\lambda_{max}$ | $\Delta\lambda$ | $\lambda_{max}$ | $\Delta\lambda$ |
| Wild-type | 505 (493) | 0 (0) | 513 | 0 |
| E68D | 499 (483) | −6 (−10) | 493 | −20 |
| E68Q | 505 (493) | 0 (0) | 515 | 2 |
| D234E | 499 (481) | −6 (−12) | 503 | −10 |
| D234N (deprotonated E68) | 507 (487) | 2 (−6) | | |
| D234N (protonated E68) | 534 (517) | 29 (24) | 510 | −3 |
| E68Q/D234N | 538 (522) | 33 (29) | | |
| E68Q/D234N (with Cl⁻)† | 509 (491) | 4 (−2) | 520 | 7 |

*Average values of 10 MD-generated structures, which are finally QM/MM-optimized. Absorption wavelengths were calculated using **Equation 1** with $\Delta E_{HOMO\text{-}LUMO}$. Absorption wavelengths calculated using **Equation 2** with $E_{TD\text{-}DFT}$ are shown in parenthesis.
†With Cl⁻ binding at the Schiff base moiety (see below).

Asp234 in wild-type and protonated Glu234 in D234E *Gt*ACR1s to the absorption wavelengths are small (**Supplementary file 1D**).

As far as we are aware, the absorption wavelength of the isolated E68Q/D234N *Gt*ACR1 is not reported (e.g., **Sineshchekov et al., 2016**; **Dreier et al., 2021**). We successfully isolated a photoactive form of E68Q/D234N *Gt*ACR1 using the HEK293 cell expression system, which has been widely used for the functional expression in animal rhodopsins (**Kojima et al., 2017**; **Yamashita et al., 2010**). The experimentally measured absorption wavelength in the isolated E68Q/D234N protein is 7 nm longer than that in the wild-type protein (**Figure 3**), which indicates that Glu68 or Asp234 must be deprotonated in the wild-type *Gt*ACR1. As Asp234 is closer to the Schiff base (3.2 Å) than Glu68 (5.3 Å) (**Li et al., 2019**), it seems more likely that Asp234 is deprotonated in the wild-type *Gt*ACR1.

Microbial rhodopsins, including Chrimson and Rh-PDE (**Vierock et al., 2017**; **Watari et al., 2019**), have more than one deprotonated acidic residue adjacent to the Schiff base (**Tsujimura and Ishikita, 2020**). The loss of two acidic residues upon the E68Q/D234N mutation requires an additional negative charge as far as the Schiff base remains protonated. Thus, it seems possible that Cl⁻ exist to stabilize the protonated Schiff base specifically in the E68Q/D234N *Gt*ACR1 because the next closest acidic residue, Glu60, is 10 Å away from the Schiff base. The presence of Cl⁻ in the E68Q/D234N *Gt*ACR1 is not reported. To investigate the existence of Cl⁻, isolated E68Q/D234N samples were solubilized in Cl⁻-free buffer. However, denaturation of the samples did not allow us to conclude the existence of Cl⁻. In QM/MM calculations and MD simulations, the binding of Cl⁻ at Thr71/Asn234 or Ser97/Lys238 is more stable in the E68Q/D234N *Gt*ACR1 than in the wild-type protein (**Figure 4—figure supplement 2**, **Supplementary file 1E**). The increase in the calculated absorption wavelength upon the E68Q/D234N mutation (33 nm) is overestimated in the absence of Cl⁻, whereas the corresponding increase (4 nm) is at the same level as that measured experimentally in the presence of Cl⁻ (**Table 4**). Thus, Cl⁻ is likely to exist near the protonated Schiff base to compensate for the loss of two acidic residues in the E68Q/D234N *Gt*ACR1.

The E68Q mutation does not alter the absorption wavelength (**Table 4**) as reported previously (**Sineshchekov et al., 2016**; **Yi et al., 2016**), thereby suggesting that Glu68 is protonated in the presence of deprotonated Asp234 (e.g., wild-type *Gt*ACR1) (**Table 1**).

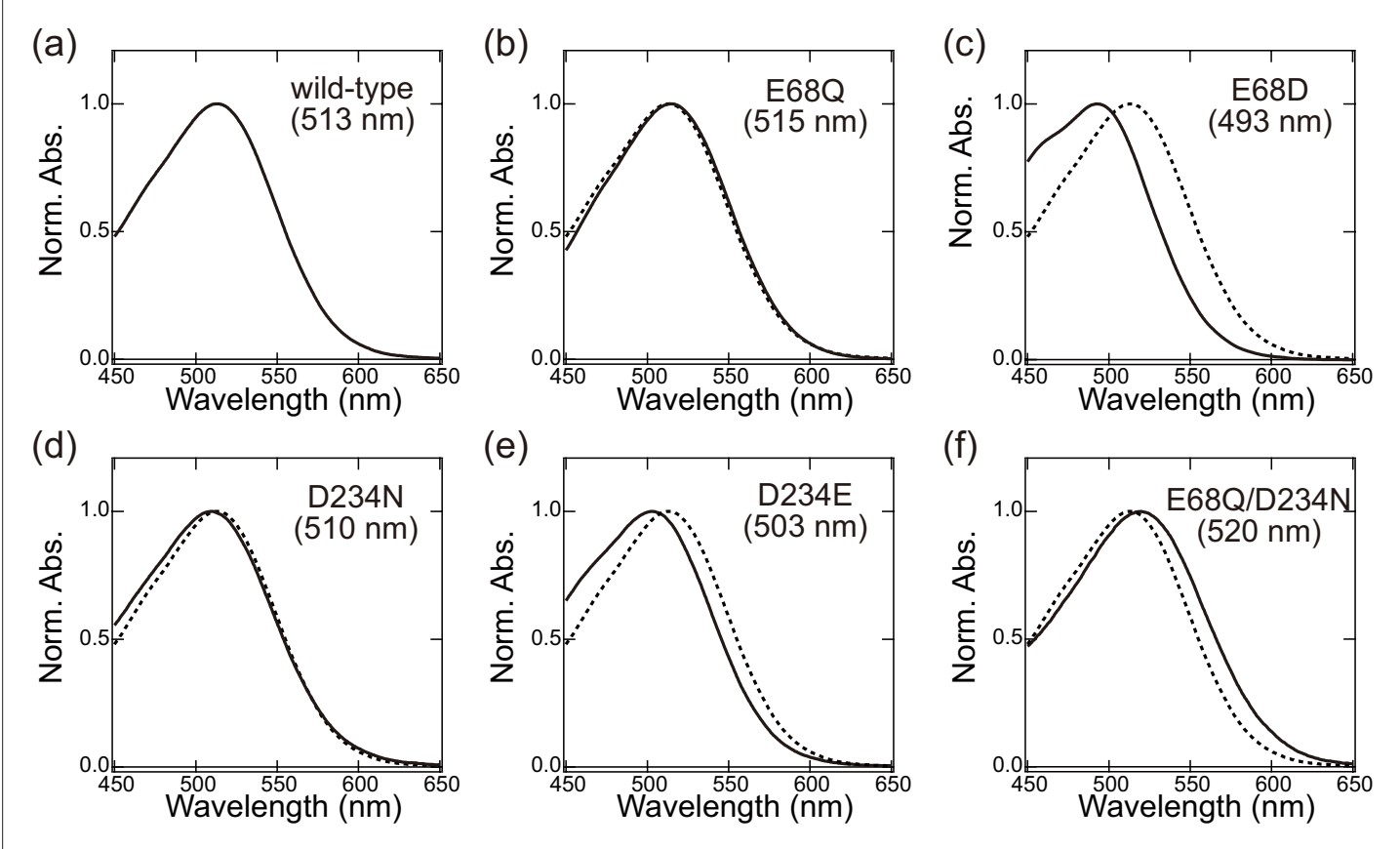

**Figure 3.** Absorption spectrum of (**a**) wild-type, (**b**) E68Q, (**c**) E68D, (**d**) D234N, (**e**) D234E, and (**f**) E68Q/D234N *Gt*ACR1s in DDM micelles. Spectra are normalized at peak absorbance. The spectra for the wild-type (dotted curves in panels b–f) are shown for comparison. For each absorption spectrum, the results of four-traces were averaged to improve the signal-to-noise ratio. Absorption wavelengths were measured from the averaged spectra.

The online version of this article includes the following figure supplement(s) for figure 3:

**Figure supplement 1.** Absorption wavelengths of wild-type and mutant *Gt*ACR1s calculated using (**a**) *Equation 1* with $\Delta E_{HOMO\text{-}LUMO}$ and (**b**) *Equation 2* with $E_{TD\text{-}DFT}$.

**Figure supplement 2.** Absorption wavelengths of wild-type and mutant *Gt*ACR1s.

In general, blue light-sensitive microbial rhodopsins (e.g., Sensory rhodopsin II and *Chlamydomonas* channelrhodopsins) show the main absorbance peak with spectral shoulder at shorter wavelength region (e.g., *Takahashi et al., 1990*). Based on these, it seems likely that the wide band of E68D is due to the existence of the spectral shoulder of this blue-shifted mutant.

## Discussion

Our finding of deprotonated Asp234 in the ground state of *Gt*ACR1 can explain the following observations: loss of photocurrent upon the D234N mutation (*Kim et al., 2018*) can be due to loss of Asp234, which is deprotonated in the wild-type *Gt*ACR1. It seems possible that the electrostatic interaction between deprotonated Asp234 and channel-gating Arg94 (*Li et al., 2021*; *Table 2*) is absent in the D234N *Gt*ACR1, leading to loss of the photocurrent (*Kim et al., 2018*). Intriguingly, MD simulations show that upon the D234N mutation, deprotonated Glu68 reorients toward and interferes with the channel bottle neck (*Figure 5*). It seems likely that Glu68 acts as a proton acceptor, forming the M-state (i.e., fast channel closing), in the D234N *Gt*ACR1 (*Sineshchekov et al., 2016*), as deprotonated Glu68 is sufficiently close to the protonated Schiff base (*Figure 4b*). The reorientation of deprotonated negatively charged Glu68 toward the protonated Schiff base and the interference with the channel bottle neck may also explain why the photocurrent owing to the anion conduction

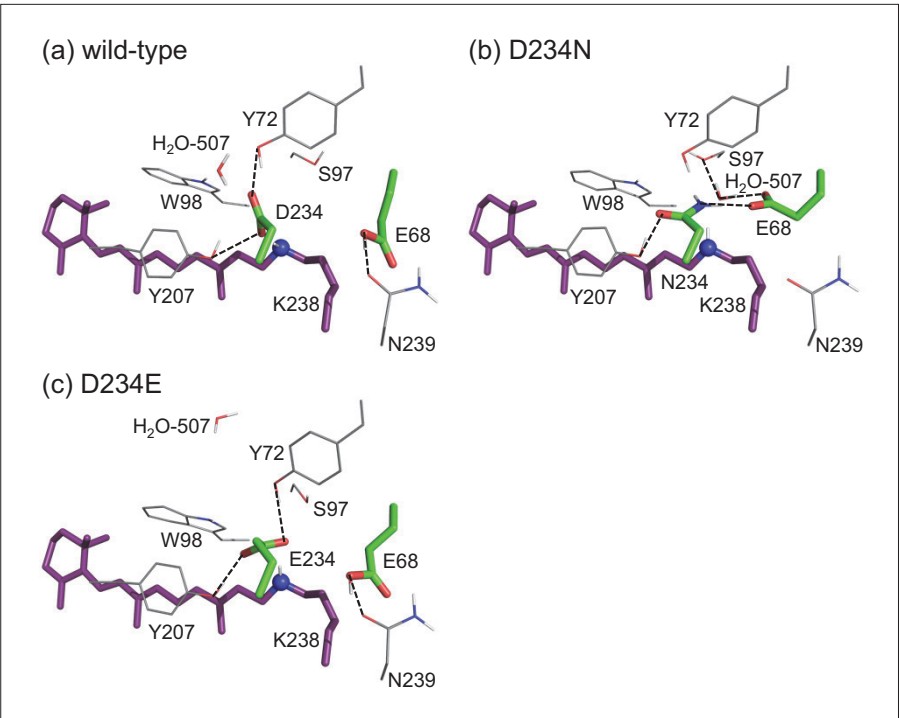

**Figure 4.** The H-bond network of the Schiff base of (**a**) wild-type, (**b**) D234N, and (**c**) D234E *Gt*ACR1s. Representative MDgenerated conformations, whose absorption wavelengths are closest to the average value for the 10 MD-generated structures, are shown. Dotted lines indicate H-bonds.

The online version of this article includes the following figure supplement(s) for figure 4:

**Figure supplement 1.** The distances between Schiff base and Asp234 in the wild-type *Gt*ACR1 and between Schiff base and Glu234 in the D234E *Gt*ACR1 during the 1 ns production run.

**Figure supplement 2.** QM/MM-optimized Cl⁻-binding structures of (**a**) wild-type and (**b**) E68Q/D234N mutant *Gt*ACR1s.

is abolished (**Kim et al., 2018**) irrespective of the accumulation of the M-state (**Sineshchekov et al., 2016**) (i.e., with deprotonated Schiff base) in the D234N *Gt*ACR1. It seems likely that the anion conduction is inhibited in the anion conducting L-state in the D234N *Gt*ACR1 because Glu68 already interferes with the channel bottle neck in the ground state. Thus, not only the gating (Arg94) but also the conduction (Glu68) can be inhibited in the D234N *Gt*ACR1.

The mechanism is also likely to hold true for the M-state in the wild-type *Gt*ACR1, although the MD simulations were conducted based on the dark state structures. As far as we are aware, no intermediate structures of *Gt*ACR1 have been reported. The recent time-resolved X-ray free electron laser (XFEL) structures of cation channel-rhodopsin C1C2 show that the distance between the Schiff base and Glu129 (Glu68 in *Gt*ACR1) remains unaffected during the early part of the photocycle irrespective of the isomerization of the retinal (**Figure 5—figure supplement 1**; **Oda et al., 2021**). In C1C2, not only the isomerization of the retinal but also a protein conformational change is required for the conducting-channel formation during the photocycle, as no continuous channel exists in the ground state (**Kato et al., 2012**). In contrast, a continuous channel spanning through the protein already exists in the ground

**Table 5.** Components that contribute to the absorption wavelength in the wild-type and D234N *Gt*ACR1s (nm).

The contributions were analyzed using a single MD-generated structure whose absorption wavelength is closest to the average value of the 10 MD-generated structures.

|  | Wild-type | D234N |
|---|---|---|
| Glu68 | 0 | −20 |
| Asp234 | −28 | |
| Asn234 | | 0 |
| H₂O | 0 | −7 |
| Total | −28 | −27 |

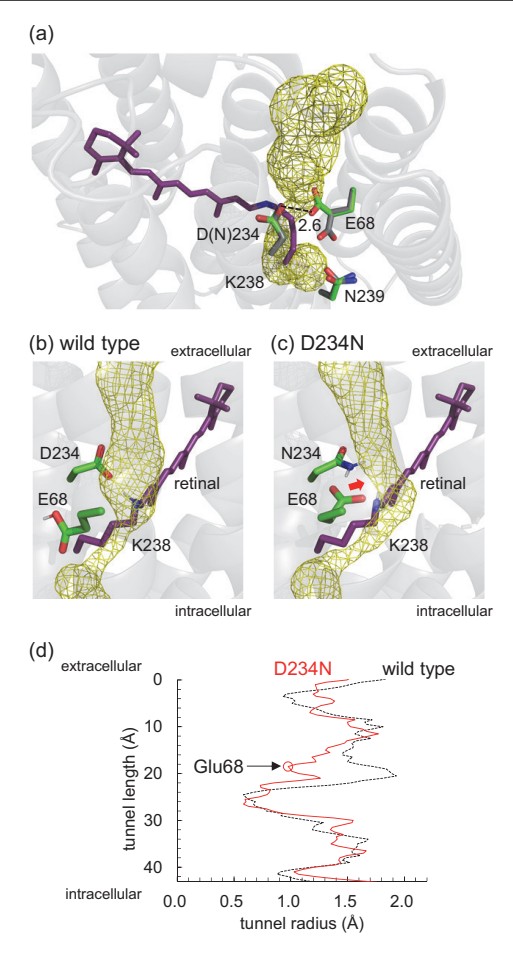

**Figure 5.** Channel space in *Gt*ACR1. (**a**) Channel space and side-chain orientations in the wild-type (gray sticks, PDB code 6EDQ; *Li et al., 2019*) and D234N (green sticks, a representative MD-generated conformation) *Gt*ACR1s. The yellow mesh indicates the channel space in the wild-type *Gt*ACR1 analyzed using the CAVER program (*Chovancova et al., 2012*). Note that the channel space is consistent with that reported by *Li et al., 2019*. Channel space and side-chain orientations in the representative MD-generated structures of (**b**) wild-type and (**c**) D234N *Gt*ACR1s. *Chovancova et al., 2012* The red arrow indicates the decrease in the channel space (radius) owing to the approach of Glu68. (**d**) Channel radii along the channel in wild-type (dotted black line) and D234N (solid red line) *Gt*ACR1s. The red open circle indicates the constriction created by reoriented Glu68 in D234N *Gt*ACR1.

The online version of this article includes the following figure supplement(s) for figure 5:

**Figure supplement 1.** Retinal Schiff base, Glu129 (Glu68 in *Gt*ACR1), and Asp292 (Asp234 in *Gt*ACR1) in the structures of the ground state (green sticks, PDB code 7C86; *Oda et al., 2021*) and 4 ms after light irradiation (cyan sticks, PDB code 7E6X; *Oda et al., 2021*) of cation channelrhodopsin C1C2.

state of *Gt*ACR1 (*Figure 5*; *Li et al., 2019*). From the analogy, it seems plausible that the Schiff base interacts electrostatically with Glu68 in the M-state, inhibiting the anion conduction.

The formation of an H-bond between Asn234 and deprotonated Glu68 in the D234N *Gt*ACR1 (*Figure 5*) also suggests that Glu68 accepts the proton from protonated Asp234 in the M-state of the wild-type *Gt*ACR1 (*Figure 6*). Based on the observation of the ground state structure, it seems possible that the proton transfer pathway that proceeds from the protonated Schiff base via deprotonated Asp234 toward protonated Glu68 can also form in the M-state. Protonated Glu68 can accept the proton from transiently protonated Asp234 and simultaneously donates the proton to the adjacent acceptor group in the H-bond network e.g., Asp-L213 in the bacterial photosynthetic reaction center (*Sugo et al., 2021*).

Then, the absence of Glu68 as a proton acceptor of Asp234 may affect the release of the proton from the Schiff base in the E68Q *Gt*ACR1. Indeed, it has been reported that the E68Q mutation affects the channel-gating mechanism (*Sineshchekov et al., 2015*), leading to a decrease in the M-state (i.e., deprotonated Schiff base) accumulation (*Sineshchekov et al., 2016*). The conformation of Glu68 as a proton acceptor of Asp234 interferes with the channel bottle neck (*Figure 5*). This may explain why the M-state formation (i.e., release of the proton from the Schiff base via Asp234 to Glu68) corresponds to the fast channel closing (*Figures 2 and 6*). This may also explain why the mutation of Glu68 to glutamine leads to a suppression of the fast channel closing at a physiological pH (*Sineshchekov et al., 2015*).

## Conclusions

It was proposed that Asp234 was protonated in the wild-type *Gt*ACR1 (*Kim et al., 2018*; *Sineshchekov et al., 2016*; *Yi et al., 2016*; *Kandori, 2020*) from the following results: (i) the absorption wavelength remains unchanged upon the D234N mutation (*Kim et al., 2018*; *Sineshchekov et al., 2016*; *Yi et al., 2016*); (ii) the C=C stretching frequency of the retinal is not significantly affected upon the D234N mutation in resonance Raman spectroscopy, which implies that the electrostatic interaction between the retinal and the protein environment remains unchanged (*Yi et al., 2016*); and (iii) the C=O stretching frequencies of 1740 (−)/1732 (+) cm⁻¹ for a protonated carboxylate, which is observed in the wild-type *Gt*ACR1, disappear in the D234N *Gt*ACR1 at 77 K (*Kim et al., 2018*). However, the C=O stretching frequencies

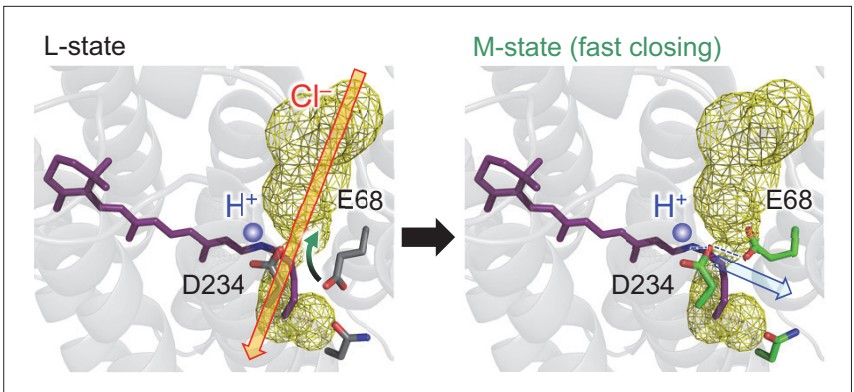

**Figure 6.** Formation of the proton transfer pathway and disconnection of the anion conduction channel during the L- to M-state transition. The channel is open (red open arrow) in the anion conducting L-state. The reorientation of the Glu68 side chain (black and green curved arrow) leads to the formation of the H-bond network that proceeds from the Schiff base to Glu68 via Asp234 (blue dotted lines) and the closing of the channel bottle neck (fast channel closing). Thus, the release of the proton from the Schiff base toward Glu68 occurs in the M-state (blue open arrow).

of 1740 (–)/1732 (+) cm$^{-1}$ for protonated Asp234 at 77 K were not observed at 293 K by *Dreier et al., 2021*.

In contrast, the present results show that Asp234 is deprotonated in the wild-type *Gt*ACR1, as indicated by the following findings. (i) The E68Q/D234N mutation leads to an increase in the absorption wavelength (*Table 4*), which indicates that Glu68 or Asp234 is deprotonated in the wild-type *Gt*ACR1 (*Table 1*). (ii) The absorption wavelength in the D234E *Gt*ACR1 is shorter than in the wild-type protein (*Table 4*), which can be explained only by the presence of a deprotonated acidic residue (*Table 1*, *Supplementary file 1D*). (iii) The calculated p$K_a$ value of –5 for Asp234 is lower than that of –2 for the corresponding residue Asp212 in BR (*Saito et al., 2012*; *Table 2*). The significantly low p$K_a$ value can be understood as Asp85 in BR, which increases p$K_a$(Asp212) by 6 (*Saito et al., 2012*), being replaced with Ser97 in *Gt*ACR1 (*Table 2*). The calculated protonation pattern shows that Asp234 is deprotonated in the wild-type *Gt*ACR1 even using the MD-generated conformations with protonated Asp234 (*Table 1*). (iv) Glu68, which is protonated in the wild-type *Gt*ACR1, is deprotonated in the D234N *Gt*ACR1 (*Table 1*). If Glu68 remained protonated in the D234N *Gt*ACR1, the absorption wavelength would be significantly longer as compared with the wild-type *Gt*ACR1 (*Table 4*). This is consistent with the FTIR measurements, which show that Glu68 is deprotonated in the D234N *Gt*ACR1 (*Dreier et al., 2021*). In any *Gt*ACR1, a negative charge needs to be present as far as the Schiff base is protonated. (v) Mutation of deprotonated Asp234 to uncharged asparagine does not alter the calculated absorption wavelength because deprotonated Glu68 reorients and interacts with the Schiff base in the D234N *Gt*ACR1, compensating for the change in the charge at the 234 site (*Table 4*, *Table 5*, *Figure 4*). Thus, the absence of changes in the absorption wavelength upon the D234N mutation (*Kim et al., 2018*; *Sineshchekov et al., 2016*; *Yi et al., 2016*) does not serve as a basis of the presence of protonated Asp234 in the wild-type *Gt*ACR1. The charge compensation by Glu68 can also explain why the C=C stretching frequency of the retinal, which reflects the electrostatic interaction between the retinal and the protein environment, does not significantly change upon the D234N mutation in resonance Raman spectroscopy (*Yi et al., 2016*).

The following mechanism can be deduced from the present findings: in D234N *Gt*ACR1, anionic Glu68 reorients toward the Schiff base to interact electrostatically. If Asp234 accepts a proton from the Schiff base in the M-state of the wild-type *Gt*ACR1, Glu68 is likely to reorient toward the channel, decreasing in the channel radius and inhibiting the anion conduction *structurally*. Simultaneously, the approach of anionic Glu68 toward the channel pore inhibits anion conduction *electrostatically* (*Figure 5*). The mechanism presented here explains why (i) the loss of photocurrent occurs upon the D234N mutation (*Kim et al., 2018*), (ii) the M-state formation corresponds to the fast channel closing (*Sineshchekov et al., 2016*), and (iii) the Glu68 to Gln mutation leads to a suppression of the fast channel closing at a physiological pH (*Sineshchekov et al., 2015*). The formation of the proton

transfer pathway in the M-state, which proceeds from the Schiff base via Asp234 and Glu68 toward the protein bulk surface (*Figure 6*), can explain (iv) the accumulation of the M-state (i.e., deprotonation of the Schiff base) in the wild-type, D234N, and E68Q *Gt*ACR1s.

When the properties of a protein (e.g., absorption wavelength) remain unchanged upon the mutation of aspartate to asparagine, one may assume that the aspartate is protonated. However, this does not hold true for the following cases: (i) when the aspartate is adjacent to the focusing site (e.g., forming an H-bond), because the H-bond character (e.g., polarity and pattern) of asparagine is not identical to that of protonated aspartate irrespective of the same net charge; (ii) when another titratable residue exists near the aspartate (e.g., Glu68 in *Gt*ACR1) and the protonation states of the two residues are linked. The present example shows that asparagine mutation is not always equivalent to protonated aspartate especially when it is directly involved in the H-bond with the focusing site.

## Materials and methods
### Coordinates and atomic partial charges
The atomic coordinates were taken from the X-ray structure of *Gt*ACR1 monomer unit 'A' (PDB code 6EDQ; *Li et al., 2019*). All crystal water molecules were included explicitly in calculations if not otherwise specified. During the optimization of hydrogen atom positions with CHARMM (*Brooks et al., 1983*), the positions of all heavy atoms were fixed, and all titratable groups (e.g., acidic and basic groups) were ionized. The Schiff base was considered protonated. Atomic partial charges of the amino acids and retinal were obtained from the CHARMM22 (*MacKerell et al., 1998*) parameter set.

### Protonation pattern
The computation was based on the electrostatic continuum model, solving the linear Poisson–Boltzmann equation with the MEAD program (*Bashford and Karplus, 1990*). The difference in electrostatic energy between the two protonation states, protonated and deprotonated, in a reference model system was calculated using a known experimentally measured p$K_a$ value (e.g., 4.0 for Asp; *Nozaki and Tanford, 1967*). The difference in the p$K_a$ value of the protein relative to the reference system was added to the known reference p$K_a$ value. The experimentally measured p$K_a$ values employed as references were 12.0 for Arg, 4.0 for Asp, 9.5 for Cys, 4.4 for Glu, 10.4 for Lys, 9.6 for Tyr, (*Nozaki and Tanford, 1967*), and 7.0 and 6.6 for the N$_\varepsilon$ and N$_\delta$ atoms of His, respectively (*Tanokura, 1983a*; *Tanokura, 1983b*; *Tanokura, 1983c*). All other titratable sites were fully equilibrated to the protonation state of the target site during titration. The dielectric constants were set to 4 inside the protein and 80 for water. All water molecules were considered implicitly. All computations were performed at 300 K, pH 7.0, and with an ionic strength of 100 mM. The linear Poisson–Boltzmann equation was solved using a three-step grid-focusing procedure at resolutions of 2.5, 1.0, and 0.3 Å. The ensemble of the protonation patterns was sampled by the Monte Carlo (MC) method with the Karlsberg program (*Rabenstein and Knapp, 2001*). The MC sampling yielded the probabilities [protonated] and [deprotonated] of the two protonation states of the molecule.

### MD simulations
The *Gt*ACR1 assembly was embedded in a lipid bilayer consisting of 258 1-palmitoyl-2-oleyl-sn-glycero-3-phosphocholine (POPC) molecules using CHARMM-GUI (*Jo et al., 2008*), and soaked in 29070–29072 TIP3P water models, and 5–7 chloride ions were added to neutralize the system using the VMD plug-ins (*Humphrey et al., 1996*). After structural optimization with position restraints on heavy atoms of the *Gt*ACR1 assembly, the system was heated from 0.1 to 300 K over 5.5 ps with time step of 0.01 fs, equilibrated at 300 K for 1 ns with time step of 0.5 fs, and annealed from 300 to 0 K over 5.5 ps with time step of 0.01 fs. The heating and annealing processes to energetically relax the positions of POPC and TIP3 water molecules were performed with time step of 0.01 fs, as done in previous studies (*Kurisaki et al., 2015a*; *Kurisaki et al., 2015b*). To avoid the influence of changes in the retinal Schiff base structure on the excitation energy, the position restraints on heavy atoms of side chains were released and MD simulations were performed; the system was heated from 0.1 K to 300 K over 5.5 ps with time step of 0.01 fs and equilibrated at 300 K for 1 ns with time step of 0.5 fs. The system was equilibrated at 300 K for 5 ns with time step of 1.0 fs, and a production run was conducted over 1 ns with 1.0 fs step for sampling of side-chain orientations. 10 conformations were sampled at 0.1 ns intervals during

the 1 ns production run. All MD simulations were conducted with the CHARMM22 (*MacKerell et al., 1998*) force field parameter set using the MD engine NAMD version 2.11 (*Phillips et al., 2005*). For MD simulations with time step of 1.0 fs, the SHAKE algorithm for hydrogen constraints was employed (*Ryckaert et al., 1977*). For temperature and pressure control, the Langevin thermostat and piston were used (*Feller et al., 1995*; *Kubo et al., 1991*).

POPC molecules have little effect on the calculated $pK_a$ values of Asp234 and Glu68 nor the absorption wavelengths (*Supplementary file 1F and G*). Using the resulting coordinates, the protonation state of the titratable residues was finally determined with the MEAD (*Bashford and Karplus, 1990*) and Karlsberg (*Rabenstein and Knapp, 2001*) programs in the absence of POPC molecules.

## QM/MM calculations

Using 10 MD-generated protein conformations, the geometry was optimized using a QM/MM approach in the absence of POPC molecules. The restricted density functional theory (DFT) method was employed with the B3LYP functional and LACVP* basis sets using the QSite (*Schrödinger, 2012*) program. The QM region was defined as the retinal and Schiff base (Lys238). All atomic coordinates were fully relaxed in the QM region, and the protonation pattern of titratable residues was implemented in the atomic partial charges of the corresponding MM region. In the MM region, the positions of H atoms were optimized using the OPLS2005 force field (*Jorgensen et al., 1996*), while the positions of the heavy atoms were fixed.

The absorption energy of microbial rhodopsins is highly correlated with the energy difference between highest occupied molecular orbital (HOMO) and lowest unoccupied molecular orbital (LUMO) ($\Delta E_{\text{HOMO-LUMO}}$) or the lowest excitation energy calculated using time-dependent (TD) DFT ($E_{\text{TD-DFT}}$) of the retinal Schiff base (*Tsujimura and Ishikita, 2020*; *Tsujimura et al., 2021*). To calculate absorption energies and corresponding wavelengths, the energy levels of HOMO and LUMO and the lowest excitation energies were calculated in the absence of POPC molecules. The absorption energy ($E_{abs}$ in eV) was calculated using the following equations, which are obtained for wild-type and six mutant *Gt*ACR1s (coefficients of determination $R^2$ = 0.93 for *Equation 1* and 0.73 for *Equation 2*; *Tsujimura et al., 2021*):

$$E_{abs} = 0.842 \, \Delta E_{\text{HOMO-LUMO}} + 0.408 \tag{1}$$

$$E_{abs} = 1.455 \, E_{\text{TD-DFT}} - 1.056 \tag{2}$$

The HOMO-LUMO energy gap and the lowest excitation energy were calculated based on 10 MD-generated/QM/MM-optimized protein conformations (see *Source data 1* for the atomic coordinates). Empirically, the correlation between the calculated and experimentally measured absorption energies is higher in $\Delta E_{\text{HOMO-LUMO}}$ than in $E_{\text{TD-DFT}}$ among 13 microbial rhodopsin crystal structures (*Tsujimura and Ishikita, 2020*) and *Gt*ACR1 mutants (*Tsujimura et al., 2021*). In the present study, we analyze the absorption wavelengths of microbial rhodopsin proteins using *Equation 1* based on the empirically corrected $E_{abs}$ (*Zhang and Musgrave, 2007*) (in eV) or the corresponding wavelength (in nm).

A QM/MM approach utilizing the polarizable continuum model (PCM) method with a dielectric constant of 78 for the bulk region, in which electrostatic and steric effects created by a protein environment were explicitly considered in the presence of bulk water, was employed. In the PCM method, the polarization points were placed on the spheres with a radius of 2.8 Å from the center of each atom to describe possible water molecules in the cavity. The radii of 2.8–3.0 Å from each atom center and the dielectric constant values of ~80 are likely to be optimal to reproduce the excitation energetics, as evaluated for the polarizable QM/MM/PCM approach (*Tamura et al., 2020*). The TD-DFT method with the B3LYP functional and 6-31G* basis sets was employed using the GAMESS program (*Schmidt et al., 1993*). The trends in the shifts of absorption wavelength with respect to wild-type *Gt*ACR1 remain unchanged when the functional/basis set is replaced (e.g., the CAM-B3LYP functional; *Yanai et al., 2004*; *Figure 3—figure supplement 2*).

The electrostatic contribution of the side chain in the MM region to the absorption wavelength of the retinal Schiff base was obtained from the shift in the HOMO-LUMO energy gap upon the removal of the atomic charges of the focusing side chain.

## Gene preparation

The cDNA of *Gt*ACR1 (GenBank accession number: KP171708) was optimized for human codon usage and fused to a C-terminal sequence encoding a hexahistidines-tag. The fusion product was inserted into the pCAGGS mammalian expression vector, as previously described (*Kojima et al., 2017*; *Kojima et al., 2020*). *Gt*ACR1 cDNAs containing mutations were constructed using the In-Fusion Cloning Kit according to the manufacturer's instructions (*Kojima et al., 2017*; *Kojima et al., 2020*).

## Protein expression and purification of GtACR1

The expression plasmids were transfected into HEK293T cells using the calcium-phosphate method (*Kojima et al., 2017*; *Kojima et al., 2020*). HEK 293T cells were a gift from Dr. Satoshi Koike (Tokyo Metropolitan Organization for Medical Research). We have confirmed that the identity has been authenticated by STR profiling and the cell lines tested negative for mycoplasma contamination. We have not used any cell lines from the list of commonly misidentified cell lines maintained by the International Cell Line Authentication Committee. After 1-day incubation, all-*trans*-retinal (final concentration = 5 µM) was added to transfected cells. After another day incubation, the cells were collected by centrifugation (7510 × *g* for 10 min) at 4°C and suspended in Buffer-A (50 mM HEPES [pH 7.0] and 140 mM NaCl). All-*trans*-retinal (final concentration = 0.31 µM) was added to the cell suspension to reconstitute the photoactive pigments by shaking rotatory for more than 12 hr at 4°C. Then, the cells were collected by centrifugation (12900 × *g* for 30 min) at 4°C and suspended in Buffer-A and solubilized in Buffer-B (20 mM HEPES [pH 7.4], 300 mM NaCl, 5% glycerol, and 1% dodecyl maltoside [DDM]). The solubilized fraction was collected by ultracentrifugation (169,800 × *g* for 20 min) at 4°C, and the supernatant was applied to a $Ni^{2+}$ affinity column to purify the pigments. After the column was washed with Buffer-C (20 mM HEPES [pH 7.4], 300 mM NaCl, 5% glycerol, 0.02% DDM, and 20 mM imidazole), the pigment was eluted with a linear gradient of imidazole by Buffer-D (20 mM HEPES [pH 7.4], 300 mM NaCl, 5% glycerol, 0.02% DDM, and 1 M imidazole). Purified samples were concentrated by centrifugation using an Amicon Ultra filter (30,000 $M_w$ cut-off; Millipore, USA) and the buffer was exchanged using PD-10 column (GE Healthcare, USA) to Buffer-E (20 mM HEPES [pH 7.4], 300 mM NaCl, 5% glycerol, and 0.02% DDM).

## Spectroscopic analysis

Absorption spectra of the purified proteins were recorded with a UV–visible spectrophotometer (Shimadzu, UV-2450, UV-2600) in Buffer-E. The samples were kept at 15°C using a thermostat.

## Acknowledgements

This research was supported by AMED (20dm0207060h0004) to YS and JST CREST (JPMJCR1656 to YS and HI), JSPS KAKENHI (JP21K15054 to KK, JP20K21482, JP21H0040413, and JP21H0244613 to YS, and JP18H05155, JP18H01937, JP20H03217, and JP20H05090 to HI), and the Interdisciplinary Computational Science Program in CCS, University of Tsukuba.

## Additional information

### Funding

| Funder | Grant reference number | Author |
|---|---|---|
| Japan Agency for Medical Research and Development | 20dm0207060h0004 | Yuki Sudo |
| Core Research for Evolutional Science and Technology | JPMJCR1656 | Yuki Sudo Hiroshi Ishikita |
| Japan Society for the Promotion of Science | JP21K15054 | Keiichi Kojima |

| Funder | Grant reference number | Author |
|---|---|---|
| Japan Society for the Promotion of Science | JP20K21482 | Yuki Sudo |
| Japan Society for the Promotion of Science | JP21H0040413 | Yuki Sudo |
| Japan Society for the Promotion of Science | JP21H0244613 | Yuki Sudo |
| Japan Society for the Promotion of Science | JP18H05155 | Hiroshi Ishikita |
| Japan Society for the Promotion of Science | JP18H01937 | Hiroshi Ishikita |
| Japan Society for the Promotion of Science | JP20H03217 | Hiroshi Ishikita |
| Japan Society for the Promotion of Science | JP20H05090 | Hiroshi Ishikita |
| University of Tsukuba | Interdisciplinary Computational Science Program in CCS | Hiroshi Ishikita |

The funders had no role in study design, data collection and interpretation, or the decision to submit the work for publication.

## Author contributions

Masaki Tsujimura, Investigation, Writing – original draft; Keiichi Kojima, Shiho Kawanishi, Yuki Sudo, Investigation; Hiroshi Ishikita, Investigation, Supervision, Writing – original draft, Writing - review and editing

## Author ORCIDs

Masaki Tsujimura (iD) http://orcid.org/0000-0002-7554-7372
Keiichi Kojima (iD) http://orcid.org/0000-0003-4729-0511
Yuki Sudo (iD) http://orcid.org/0000-0001-8155-9356
Hiroshi Ishikita (iD) http://orcid.org/0000-0002-5849-8150

## Decision letter and Author response

Decision letter https://doi.org/10.7554/eLife.72264.sa1
Author response https://doi.org/10.7554/eLife.72264.sa2

# Additional files

## Supplementary files

• Supplementary file 1. Calculated p*K*a, absorption wavelength, and binding energy values. (A) Calculated $pK_a$ values of Glu68 and Asp234 (PDB codes 6EDQ [*Li et al., 2019*] and 6CSM [*Kim et al., 2018*]). (B) Calculated $pK_a$ values of Glu68 and Asp234 in wild-type *Gt*ACR1 using the 10 MD-generated conformations with deprotonated Glu68/protonated Asp234. (C) Absorption wavelengths of 10 MD-generated structures calculated using *Equation 1* with $\Delta E_{HOMO-LUMO}$ ($\lambda_{HOMO-LUMO}$) and *Equation 2* with $E_{TD-DFT}$ ($\lambda_{TD-DFT}$) (nm). (D) Calculated absorption wavelengths of wild-type *Gt*ACR1 with protonated Asp234 and D234E *Gt*ACR1 with protonated Glu234 (nm). The absorption wavelengths were calculated using *Equation 1* with $\Delta E_{HOMO-LUMO}$ ($\lambda_{HOMO-LUMO}$) and *Equation 2* with $E_{TD-DFT}$ ($\lambda_{TD-DFT}$). $\Delta \lambda_{234}$ (nm) denotes the electrostatic contributions of protonated Asp234 in wild-type *Gt*ACR1 and protonated Glu234 in D234E *Gt*ACR1 to the absorption wavelengths. (E) Binding energies between $Cl^-$ and the surrounding environments in $Cl^-$-binding wild-type and E68Q/D234N mutant structures (kcal/mol). The binding energies were calculated based on the QM/MM-optimized and MD-based structures, using the MOE program (*Molecular Operating Environment (MOE), 2021*). (F) Calculated $pK_a$ values of Glu68 and Asp234 in wild-type *Gt*ACR1 using the 10 MD-generated conformations with protonated Glu68/deprotonated Asp234. (G) Absorption wavelengths of 10 MD-generated wild-type *Gt*ACR1 structures calculated using *Equation 1* with $\Delta E_{HOMO-LUMO}$ ($\lambda_{HOMO-LUMO}$) and *Equation 2* with $E_{TD-DFT}$ ($\lambda_{TD-DFT}$) (nm).

- Transparent reporting form
- Source data 1. Atomic coordinates of MD-generated/QM/MM-optimized wild-type and mutant *Gt*ACR1 structures.

### Data availability

All data generated or analysed during this study are included in the manuscript and supporting files.

The following previously published datasets were used:

| Author(s) | Year | Dataset title | Dataset URL | Database and Identifier |
|---|---|---|---|---|
| Li H, Huang CY, Govorunova EG, Schafer CT, Sineshchekov OA, Wang M, Zheng L, Spudich JL | 2019 | X-ray crystal structure of GtACR1 | https://www.rcsb.org/6EDQ | RCSB Protein Data Bank, 6EDQ |

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
