## [Decision Letter]

**Decision letter after peer review:**

Thank you for submitting your article "Proton-mediated gating mechanism of anion channelrhodopsin-1" for consideration by *eLife*. Your article has been reviewed by 3 peer reviewers, including Qiang Cui as the Reviewing Editor and Reviewer #1, and the evaluation has been overseen by José Faraldo-Gómez as the Senior Editor.

Essential revisions:

1) Several reviewers raised concern in using the HOMO-LUMO gap to model the absorption maximum instead of state-of-the-art QM/MM methods based on, for example, TD-DFT or ab initio methods. This should be possible for the current system considering the modest size of the QM region.

2) Essentially all reviewers are concerned about the statistical errors associated with the reported simulation data. These should be analyzed carefully and reported explicitly.

3) One reviewer highlighted the importance of examining an alternative protonation state of the system [E68(-)/D234(H)].

4) As one reviewer noted, the title of the manuscript implies insight into the "gating mechanism" of GtACR1, but it was actually never simulated like, for example, in the work from Ardevol and Hummer https://www.pnas.org/content/115/14/3557. Instead the conclusions about the gating mechanism are based on simulations of the resting state of the protein. You explained that the actual channel function is correlated with later intermediates of the photocycle (Figure 2 in the manuscript). However, already in the first intermediate the retinal changes the conformation thus the protein conformation might change, which could impact the distance to E68. Therefore, it is important to discuss the mechanistic implication of using the ground state structure in the computational analysis.

*Reviewer #1 (Recommendations for the authors):*

Overall, the authors did an excellent job in combining different computational methodologies to address the issue of residue titration states in proteins.

*Reviewer #2 (Recommendations for the authors):*

1) One of the major concerns is that the authors use HOMO-LUMO energy difference to compare them to experimental absorption maxima. A HOMO-LUMO energy gap is not an excitation energy and not the absorption maximum. The main conclusions in this paper are based on these calculations. Therefore, I am concerned about the reliability of the results. To improve it, there is a wide range of methods that can be used instead. I don't mind if the authors use TD-DFT or wave function methods as long as excitation energies are computed.

2) The authors report that 10 geometries were used for HOMO-LUMO gap computation and then averaged. However, the numbers are not shown anywhere. I would like to ask to list it in the SI. It would be for example interesting to see the fluctuation of the values because the differences between the different mutants are rather small.

3) The discussion of the gating mechanism is done on the basis of the dark state structure (Figure 5 and 6), i.e. the retinal is not isomerized and the protein conformation didn't change. So there is some uncertainty in the prediction of the gating mechanism which should be mentioned and discussed.

4) There is very little or no data from the MD trajectories which are used to draw conclusions about the protonation states. For example the RMSD from each trajectory as well as critical distances. This can be also added to the SI.

5) The main conclusion is that Asp234 is deprotonated because the D234E mutant is blue shifted. But this can also be explained by the ability of the protonated D234 to form hydrogen bonds with the Schiff base. Then the closer proximity of E234 would also result in a blue shift.

6) Also the fact that D234E/E68D double mutants is red shifted by only 7 nm can explained by the fact that in D234E the residue E68 is closer to the Schiff base but in the double mutant both are neutral. In addition, the shift is of 7 nm is rather small.

7) In several places the authors mention the similarity between the electrostatic potential in bR and GtACR1. However, bR has a second counter ion (D85) and GtACR1 has a neutral residue in this position. This is close to the Schiff bas region and is therefore important. So I recommend to rethink this phrasing.

8) How can E68 accept a proton in the wild type if it is protonated?

*Reviewer #3 (Recommendations for the authors):*

This is interesting work and I support its publication following revision. My concerns and suggestions for improvement are:

1) It is clear that one of the acidic residues, E68 or D234, must be deprotonated in order for the Schiff base to be protonated. However, the paper would be improved by a more thorough comparison of the interplay or exchangeability of E68 and D234. In many of the analyses, the authors only compare E68(H)/D234(-) to E68(H)/D234(H). They do not test E68(-)/D234(H)-e.g., MD simulations in Table 1, pKa values in tables 2 and 3, and the calculated spectroscopic shifts in table 4. The mutant results for D234N already suggest that E68 can take D234's role in stabilizing the protonated Schiff base. However, this mutant should not take the place of the wildtype system with E68(-)/D234(H). As emphasized in their concluding paragraph, Asp and Asn can behave differently.

2) For example, focusing on the measured shifts in absorption wavelengths (Table 4), mutating either residue to the opposite acidic residue (i.e., E68D and D234E) results in a decrease in wavelength. Similarly, mutating either to neutral (E68Q and D234N) results in very little shift (+2 and -3, respectively). Is this difference significant or within error of being equivalent? If the latter, why is it that the authors conclude only E68(H)/D234(-) is present in the room temperature wild type system? Couldn't E68(-)/D234(H) be a contributing species? Similarly, the authors could have calculated the shifts for E68Q in both the protonated and deprotonated forms of D234 (Table 4).

3) Focusing on pKa calculations, which will clearly hold their trends for the crystal structure, the pKa of E68 should also be measured post simulation and relaxation of the E68(-)/D234(H) state.

4) An additional concern is the lack of reported error bars. The magnitude of error should be reported for each quantitative calculated and measured result. This is essential to discern which conclusions are statistically supported and outside of their measured or calculated error bars.

5) It would help the reader to clearly delineate what is new in this work and what has been previously reported. It seems only the absorption of E68Q/D234N is new? What conclusions are new?

6) The final concluding paragraph about the influence of Asp to Asn mutations does not seem well supported by these findings. In this presented case there is a functional change due to the D234N mutation, even though there is little change in absorption due to compensating D68 deprotonation. This conclusion should perhaps be reconsidered/reframed. If kept as is, the motivation based on the reported findings needs to be more clearly explained.

7) The DFT used in the QM/MM was fairly basic (B3LYP with 6-31G* basis sets). How was this choice justified for the given system? I.e., to what higher level theory and larger basis sets was this compared to in order to verify it works well enough for these calculations?

8) Line 311 on page 17 replace ', which are caused by…asparagine,.…' with 'that are caused by.… asparagine.…' removing the commas since the clause specifies the subject-it is not additional information about the subject.

[Editors' note: further revisions were suggested prior to acceptance, as described below.]

Thank you for resubmitting your work entitled "Proton transfer pathway in anion channelrhodopsin-1" for further consideration by *eLife*. Your revised article has been reviewed by 3 peer reviewers, one of whom is a member of our Board of Reviewing Editors, and the evaluation has been overseen by José Faraldo-Gómez as the Senior Editor.

Editors and reviewers concur that the manuscript has been improved; however, there are some remaining issues that must be addressed or clarified before the manuscript can be accepted for publication, especially about the TD-DFT calculations and MD simulations for the pKa evaluations. We urge you to provide a compelling resolution to these issues.

*Reviewer #1:*

The authors included additional TD-DFT calculations, and also sampled more configurations as well as protonation patterns. They have also clarified a few questions raised in the last round of review. In my opinion, the revision is acceptable for publication in *eLife*.

*Reviewer #2:*

1) The authors have provided TD-DFT results, following the suggestion from all three reviewers. The results are qualitatively similar to the HOMO-LUMO energy gaps. But there is one important difference. The structure with Cl^-^ is blue shifted with respect to the parent state at the TD-DFT level, while it is red shifted when the HOMO-LUMO energy gap is used for the calculation of excitation energies.

A comparison of the differences in orbital energies to absorption maxima is physically incorrect (also pointed out by other reviewers). Therefore, I recommend the authors to add the TD-DFT results to the table 4 in the manuscript.

2) I appreciate adding the excitation energies from the 10 snapshots of each GtACR1 variant. It shows that the wide distribution for E68D agrees with the wide band from experiment. Maybe this is due to some heterogeneity in the structure. It would be interesting to check this.

3) It is not clear how the authors know that the M intermediate is the conducting intermediate in GtACR1. As far as I know the structure of the M intermediate is also not known for C1C2, the artificial chimaera of channelrhodopsin 1 and 2. More importantly, it is not just the isomerization but also a conformational change in the protein which allows the pore formation. It is not clear how this is taken into account in the simulation. I'm not asking the author to redo the simulations, but simply inform the reader about this important differences.

4) The proton transfer is outside of my expertise. I thought the Grotthuss mechanism is proposed for a network of water molecules.

In addition to these replies, the use of a 0.01 fs time step for the numerical integration of the equations of motions in MD makes no sense. This is at least 50 times too small.

*Reviewer #3:*

My one remaining concern is that the length of the MD simulations in each protonation state should be stated and the method for selecting conformations from those simulations should be described. The reason for this is that the pKa calculations will be most dependent on conformation. Their conclusions hinge on (1) having run simulations long enough to stabilize each charge state, and (2) selecting a reasonable number of conformations that were representative of the stablilized ensemble. Ideally they would report the number of conformations they chose, how probable those structures were in the simulations (cluster and count based on positions of the important residues), and show a representative figure of the structures in the SI.

Other than this, I am happy with the author's revisions.

---

## [Author Response]

Essential revisions:1) Several reviewers raised concern in using the HOMO-LUMO gap to model the absorption maximum instead of state-of-the-art QM/MM methods based on, for example, TD-DFT or ab initio methods. This should be possible for the current system considering the modest size of the QM region.

We have calculated in all cases using (i) TD-DFT and (ii) different basis set and functional accordingly (Supplementary files 1C, D, and G, Figure 3—figure supplement 1 and 2).

2) Essentially all reviewers are concerned about the statistical errors associated with the reported simulation data. These should be analyzed carefully and reported explicitly.

We have shown the statistical error in Figure 3—figure supplement 1 and 2.

3) One reviewer highlighted the importance of examining an alternative protonation state of the system [E68(-)/D234(H)].

According to the suggestion from reviewer 3 (points 1, 2, and 3), we have conducted MD simulations assuming the E68(-)/D234(H) state and repeated electrostatic calculations for the protonation pattern, solving the Poisson-Boltzmann equation. The results have been added mainly to Table 1.

4) As one reviewer noted, the title of the manuscript implies insight into the "gating mechanism" of GtACR1, but it was actually never simulated like, for example, in the work from Ardevol and Hummer https://www.pnas.org/content/115/14/3557. Instead the conclusions about the gating mechanism are based on simulations of the resting state of the protein. You explained that the actual channel function is correlated with later intermediates of the photocycle (Figure 2 in the manuscript). However, already in the first intermediate the retinal changes the conformation thus the protein conformation might change, which could impact the distance to E68. Therefore, it is important to discuss the mechanistic implication of using the ground state structure in the computational analysis.

As the editor pointed out, we discussed only based on the ground state structure. We realized that the main focus is on the protonation states and proton transfer, rather than gating mechanism.

Thus, we have modified the title of the manuscript as follows: “Proton transfer pathway in anion channelrhodopsin-1”.

Reviewer #2 (Recommendations for the authors):1) One of the major concerns is that the authors use HOMO-LUMO energy difference to compare them to experimental absorption maxima. A HOMO-LUMO energy gap is not an excitation energy and not the absorption maximum. The main conclusions in this paper are based on these calculations. Therefore, I am concerned about the reliability of the results. To improve it, there is a wide range of methods that can be used instead. I don't mind if the authors use TD-DFT or wave function methods as long as excitation energies are computed.

The comment is correct. We have also shown the results of TD-DFT. The two results are essentially the same (Figure 3—figure supplement 1 and 2 and Supplementary file 1C). Just empirically, the correlation between the experimentally measured absorption and calculated wavelengths is higher in ∆*E*_HOMO-LUMO_ than in *E*_TD-DFT_ among 13 microbial rhodopsin crystal structures (J. Phys. Chem. B 124 (2020) 11819) and *Gt*ACR1 mutants (Biochim. Biophys. Acta 1862 (2021) 148349), although the reason is unclear.

2) The authors report that 10 geometries were used for HOMO-LUMO gap computation and then averaged. However, the numbers are not shown anywhere. I would like to ask to list it in the SI. It would be for example interesting to see the fluctuation of the values because the differences between the different mutants are rather small.

We have shown all values for the 10 geometries of all mutant *Gt*ACR1s in Supplementary file 1C. To show the fluctuation of each value with error bar, we have also made Figure 3—figure supplement 1.

3) The discussion of the gating mechanism is done on the basis of the dark state structure (Figure 5 and 6), i.e. the retinal is not isomerized and the protein conformation didn't change. So there is some uncertainty in the prediction of the gating mechanism which should be mentioned and discussed.

The comment is correct. To evaluate whether the present mechanism deduced from the ground state structure is applicable to the M-state, we have analysed the structural difference between the all-*trans* and 13-*cis* conformations in the reported structures of C1C2. The most crucial distance, the Schiff base…counterion distance, is identical for the two conformations in C1C2 (7.6 Å and 7.2 Å, respectively, Figure 5—figure supplement 1). The overall geometry at the Schiff base/counterion moiety remains unchanged. From the analogy, we concluded that the present mechanism is likely to hold true for the M-state in *Gt*ACR1.

We have illustrated the structural difference in C1C2 (Figure 5—figure supplement 1) and discussed the mechanism for *Gt*ACR1 from the analogy (near Figure 5).

4) There is very little or no data from the MD trajectories which are used to draw conclusions about the protonation states. For example the RMSD from each trajectory as well as critical distances. This can be also added to the SI.

We consider N(Schiff Base)…O(Asp/Glu234) as the most critical distance in the present study, because the protonation state of Asp/Glu234 is the central focus and it affects the absorption wavelength most critically.

We have shown the changes in N…O distance in wild type and D234E *Gt*ACR1s during MD simulations (Figure 4—figure supplement 1).

5) The main conclusion is that Asp234 is deprotonated because the D234E mutant is blue shifted. But this can also be explained by the ability of the protonated D234 to form hydrogen bonds with the Schiff base. Then the closer proximity of E234 would also result in a blue shift.

As a careful check suggested by the reviewer, we have conducted MD simulations with protonated Asp/Glu234. The calculated absorption wavelength with protonated Glu234 in the D234E mutant remains unchanged with respect to wild type, which cannot explain the experimentally measured blue-shift of 10 nm. This is because when Asp/Glu234 are protonated, the electrostatic influence of charge neutral residues on the Schiff base are significantly small (with respect to deprotonated anionic Asp/Glu234), being almost the same (Supplementary file 1D).

We have summarized the calculated absorption wavelengths with protonated Asp/Glu234 in Supplementary file 1D.

6) Also the fact that D234E/E68D double mutants is red shifted by only 7 nm can explained by the fact that in D234E the residue E68 is closer to the Schiff base but in the double mutant both are neutral. In addition, the shift is of 7 nm is rather small.

The D234E/E68D mutant was not analysed in the present study. From “*red shifted by only 7 nm*”, we guess that the reviewer probably referred to E68Q/D234N mutant. In this case, however, the suggestion is unclear to us.

7) In several places the authors mention the similarity between the electrostatic potential in bR and GtACR1. However, bR has a second counter ion (D85) and GtACR1 has a neutral residue in this position. This is close to the Schiff bas region and is therefore important. So I recommend to rethink this phrasing.

The comment is correct. Asp85 in BR is replaced with Ser97 in *Gt*ACR1. We have rephrased the corresponding sentence in page 4.

8) How can E68 accept a proton in the wild type if it is protonated?

As far as protonated acidic residues are involved in a Grotthuss-like conduit, it can mediate proton transfer, accepting the proton from the donor site and simultaneously donating the proton to the acceptor site (e.g., protonated AspL213 in the proton transfer pathway toward quinone in the photosynthetic reaction center. It has long been suggested that Asp-L213 is protonated and accepts a proton from Asp-M17 in the upstream and donates the proton to Ser-L213, the direct proton donor to the quinone). We have added an explanation to Discussion.

Reviewer #3 (Recommendations for the authors):This is interesting work and I support its publication following revision. My concerns and suggestions for improvement are:1) It is clear that one of the acidic residues, E68 or D234, must be deprotonated in order for the Schiff base to be protonated. However, the paper would be improved by a more thorough comparison of the interplay or exchangeability of E68 and D234. In many of the analyses, the authors only compare E68(H)/D234(-) to E68(H)/D234(H). They do not test E68(-)/D234(H)-e.g., MD simulations in Table 1, pKa values in tables 2 and 3, and the calculated spectroscopic shifts in table 4. The mutant results for D234N already suggest that E68 can take D234's role in stabilizing the protonated Schiff base. However, this mutant should not take the place of the wildtype system with E68(-)/D234(H). As emphasized in their concluding paragraph, Asp and Asn can behave differently.

We have performed MD simulations with E68(-)/D234(H). Although the protonation state was fixed to E68(-)/D234(H) during MD simulations, the protonation pattern calculated using the MD-generated geometry indicates that Glu68 is protonated and Asp234 is deprotonated. That is, the protein electrostatic environment is already preorganized to stabilize protonated Glu68 and deprotonated Asp234, eliminating the possibility of E68(-)/D234(H). The result has been added to Table 1.

Using the 10 MD-generated conformations, we have calculated the p*K*_a_ values. All conformations show that Glu68 is protonated and Asp234 is deprotonated. The result has been summarized in Supplementary file 1B.

These results suggest that the protein electrostatic environment does not allow to have E68(-)/D234(H) at least in the original geometry of the ground-state structure.

Because the E68(-)/D234(H) state cannot be obtained in all MD-generated conformations, we were unable to calculate the corresponding absorption wavelength.

2) For example, focusing on the measured shifts in absorption wavelengths (Table 4), mutating either residue to the opposite acidic residue (i.e., E68D and D234E) results in a decrease in wavelength. Similarly, mutating either to neutral (E68Q and D234N) results in very little shift (+2 and -3, respectively). Is this difference significant or within error of being equivalent? If the latter, why is it that the authors conclude only E68(H)/D234(-) is present in the room temperature wild type system? Couldn't E68(-)/D234(H) be a contributing species? Similarly, the authors could have calculated the shifts for E68Q in both the protonated and deprotonated forms of D234 (Table 4).

E68(-)/D234(H) is not stable in the original geometry of the crystal structure (i.e., without performing MD simulations, Tables 2 and 3).

Even if we intentionally force to stabilize the E68(-)/D234(H) state at a room temperature (i.e., MD simulations, Table 1), the resulting conformation is far from having E68(-)/D234(H) (Table 1, wild type (3)). The energetically unstable E68(-)/D234(H) in any conformation indicates that electrostatic interactions between E68/D234 with the protein environment originating from the residues listed in Tables 2 and 3 cannot allow them to have E68(-)/D234(H). Since QM/MM calculations need energetically optimized geometry optimization before calculating absorption wavelength (otherwise it makes no sense), no further calculation for the energetically unstable E68(-)/D234(H) conformation can be performed.

We have added the result of E68(-)/D234(H) to Table 1.

3) Focusing on pKa calculations, which will clearly hold their trends for the crystal structure, the pKa of E68 should also be measured post simulation and relaxation of the E68(-)/D234(H) state.

We have shown the result in Table 1, wild type (3), and in Supplementary file 1B.

4) An additional concern is the lack of reported error bars. The magnitude of error should be reported for each quantitative calculated and measured result. This is essential to discern which conclusions are statistically supported and outside of their measured or calculated error bars.

We have summarized the statistical error of the reported data for calculations in Figure 3—figure supplement 1.

For each measured absorption spectrum, the results of 4-traces were averaged to improve the signal-to-noise ratio. Absorption wavelengths were measured from the averaged spectra. This was not mentioned in the original version. We have added the explanation to the caption, Figure 3.

5) It would help the reader to clearly delineate what is new in this work and what has been previously reported. It seems only the absorption of E68Q/D234N is new? What conclusions are new?

We have explained the present new findings more clearly, making the “Conclusion” chapter and described the points therein.

6) The final concluding paragraph about the influence of Asp to Asn mutations does not seem well supported by these findings. In this presented case there is a functional change due to the D234N mutation, even though there is little change in absorption due to compensating D68 deprotonation. This conclusion should perhaps be reconsidered/reframed. If kept as is, the motivation based on the reported findings needs to be more clearly explained.

The original sentence was unclear to readers. We have rephrased the paragraph entirely, as we consider that this message is the most important outcome of the present study (that is, “Asp234 is deprotonated even though the absorption wavelength remained unchanged upon the D234N mutation”) and should be presented appropriately.

7) The DFT used in the QM/MM was fairly basic (B3LYP with 6-31G* basis sets). How was this choice justified for the given system? I.e., to what higher level theory and larger basis sets was this compared to in order to verify it works well enough for these calculations?

We have also calculated with (i) 6-31G** (basis set) and (ii) CAM-B3LYP (functional). The trends in the shifts of absorption wavelength with respect to wild type *Gt*ACR1 remain unchanged. The results have been shown in Figure 3—figure supplement 2.

8) Line 311 on page 17 replace ', which are caused by…asparagine,.…' with 'that are caused by.… asparagine.…' removing the commas since the clause specifies the subject-it is not additional information about the subject.

Thank you very much for pointing out our grammatical error. The corresponding sentence has been removed to address the point 6.

[Editors' note: further revisions were suggested prior to acceptance, as described below.]

Editors and reviewers concur that the manuscript has been improved; however, there are some remaining issues that must be addressed or clarified before the manuscript can be accepted for publication, especially about the TD-DFT calculations and MD simulations for the pKa evaluations. We urge you to provide a compelling resolution to these issues.Reviewer #2:1) The authors have provided TD-DFT results, following the suggestion from all three reviewers. The results are qualitatively similar to the HOMO-LUMO energy gaps. But there is one important difference. The structure with Cl^-^ is blue shifted with respect to the parent state at the TD-DFT level, while it is red shifted when the HOMO-LUMO energy gap is used for the calculation of excitation energies.A comparison of the differences in orbital energies to absorption maxima is physically incorrect (also pointed out by other reviewers). Therefore, I recommend the authors to add the TD-DFT results to the table 4 in the manuscript.

Thank you very much for your helpful advice. We have added the TD-DFT results to Table 4.

2) I appreciate adding the excitation energies from the 10 snapshots of each GtACR1 variant. It shows that the wide distribution for E68D agrees with the wide band from experiment. Maybe this is due to some heterogeneity in the structure. It would be interesting to check this.

After long discussion with the members, we have concluded that this is more likely due to “spectral shoulder” than “some heterogeneity in the structure”.

We have stated in Results as follows:

“In general, blue light-sensitive microbial rhodopsins (e.g., Sensory rhodopsin II and *Chlamydomonas* channelrhodopsins) show the main absorbance peak with spectral shoulder at shorter wavelength region (e.g. Takahashi, T. et al. Biochemistry 1990, 29, 8467-8474). Based on these, it seems likely that the wide band of E68D is due to the existence of the spectral shoulder of this blue-shifted mutant.”

3) It is not clear how the authors know that the M intermediate is the conducting intermediate in GtACR1.

The M intermediate state is not the conducting intermediate. Instead, the L intermediate is the anion conducting intermediate and the fast channel closing occurs during the L to M transition, as shown in Figure 2.

As far as I know the structure of the M intermediate is also not known for C1C2, the artificial chimaera of channelrhodopsin 1 and 2.

The comment is correct. The structure shown in Figure 5—figure supplement 1 (PDB code 7E6X) is just a “putative” M-state structure (the structure at the time point of ∆*t* = 4 ms, where the M-state is accumulated in the C1C2 crystals (Oda, K. et al. *eLife* 2021, 10, e62389)).

Accordingly, removing “the M-state” from C1C2, we have corrected the main text and the legend of Figure 5—figure supplement 1 to avoid confusion.

More importantly, it is not just the isomerization but also a conformational change in the protein which allows the pore formation. It is not clear how this is taken into account in the simulation. I'm not asking the author to redo the simulations, but simply inform the reader about this important differences.

We have stated that not only the isomerization but also a conformation change is required for the conducting-channel formation during the photoycle (in Discussion).

4) The proton transfer is outside of my expertise. I thought the Grotthuss mechanism is proposed for a network of water molecules.

The comment is correct. We have replaced “Grotthuss-like proton conduit” with “H-bond network”.

In addition to these replies, the use of a 0.01 fs time step for the numerical integration of the equations of motions in MD makes no sense. This is at least 50 times too small.

The comment is indeed correct. As the reviewer suggested, the time step of 0.01 fs is too short. Although the short time step was used only for heating and annealing processes, it makes no sense. We will not use a too-short time step next time. Thank you very much for the important comment.

Reviewer #3:My one remaining concern is that the length of the MD simulations in each protonation state should be stated and the method for selecting conformations from those simulations should be described. The reason for this is that the pKa calculations will be most dependent on conformation. Their conclusions hinge on (1) having run simulations long enough to stabilize each charge state, and (2) selecting a reasonable number of conformations that were representative of the stablilized ensemble.

We have shown the length of the MD simulations and the method for selecting conformations in the Method section and the footnote of Table 1.

Ideally they would report the number of conformations they chose, how probable those structures were in the simulations (cluster and count based on positions of the important residues), and show a representative figure of the structures in the SI.

In each protonation state, deprotonated/protonated Asp234 forms H-bonds with the adjacent residues through the 1 ns product run.

We have shown the representative conformations and the distances of the H-bonds of Asp234 in Figure 2—figure supplement 1.